# DEPENDENT BIDIRECTIONAL RNN WITH EXTENDED-LONG SHORT-TERM MEMORY

## ABSTRACT

In this work, we first conduct mathematical analysis on the memory, which is defined as a function that maps an element in a sequence to the current output, of three RNN cells; namely, the simple recurrent neural network (SRN), the long short-term memory (LSTM) and the gated recurrent unit (GRU). Based on the analysis, we propose a new design, called the extended-long short-term memory (ELSTM), to extend the memory length of a cell. Next, we present a multi-task RNN model that is robust to previous erroneous predictions, called the dependent bidirectional recurrent neural network (DBRNN), for the sequence-in-sequence-out (SISO) problem. Finally, the performance of the DBRNN model with the ELSTM cell is demonstrated by experimental results.

## 1 INTRODUCTION

The recurrent neural network (RNN) has proved to be an effective solution for natural language processing (NLP) through the advancement in the last three decades (Elman, 1990; Jordan, 1997; Bayer et al., 2009; Barret & V. Le, 2016). At the cell level of a RNN, the long short-term memory (LSTM) (Hochreiter & Schmidhuber, 1997) and the gated recurrent unit (GRU) (Cho et al., 2014) are often adopted by a RNN as its low-level building cell. Being built upon these cells, various RNN models have been proposed to solve the sequence-in-sequence-out (SISO) problem. To name a few, there are the bidirectional RNN (BRNN) (Schuster & Paliwal, 1997), the encoder-decoder model (Cho et al., 2014; Sutskever et al., 2014; Vinyals et al., 2015; Bahdanau et al., 2015) and the deep RNN (Pascanu et al., 2014). Although the LSTM and the GRU were designed to enhance the memory length of RNNs and avoid the gradient vanishing/exploding issue (Hochreiter & Schmidhuber, 1997; Razvan et al., 2013; Bengio et al., 1994), a good understanding of their memory length is still lacking. Here, we define the memory of a RNN model as a function that maps an element in a sequence to current output. The first objective of this research is to analyze the memory length of three RNN cells – the simple RNN (SRN) (Elman, 1990; Jordan, 1997), the long short-term memory (LSTM) and the gated recurrent unit (GRU). This will be conducted in Sec. 2. Such analysis is different to the investigation of gradient vanishing/exploding problem in a sense that gradient vanishing/exploding problem happens during the training process, the memory analysis is, however, done on a trained RNN model. Based on the understanding from the memory analysis, we propose a new design, called the extended-long short-term memory (ELSTM), to extend the memory length of a cell in Sec.3.

As to the macro RNN model, one popular choice is the BRNN. Since the elements in BRNN output sequences should be independent of each other (Schuster & Paliwal, 1997), the BRNN cannot be used to solve dependent output sequence problem alone. Nevertheless, most language tasks do involve dependent output sequences. The second choice is the encoder-decoder system, where the attention mechanism has been introduced (Vinyals et al., 2015; Bahdanau et al., 2015) to improve its performance furthermore. As shown later in this work, the encoder-decoder system is not an efficient learner. Here, to take advantages of both the encoder-decoder and the BRNN and overcome their drawbacks, we propose a new multitask model called the dependent bidirectional recurrent neural network (DBRNN), which will be elaborated in Sec. 4. Furthermore, we conduct a series of experiments on the part of speech (POS) tagging and the dependency parsing (DP) problems in Sec. 5 to demonstrate the performance of the DBRNN model with the ELSTM cell. Finally, concluding remarks are given and future research direction is pointed out in Sec. 6.

## 2 MEMORY ANALYSIS OF SRN, LSTM AND GRU

For a large number of NLP tasks, we are concerned with finding semantic patterns from the input sequence. It was shown by Elman (1990) that the RNN builds an internal representation of semantic patterns. The memory of a cell characterizes its ability to map an input sequence of certain length into such a representation. More rigidly, we define the memory as a function that maps an element in a sequence to the current output. So the memory capability of a RNN is not only about whether an element can be mapped into current output, but also how this mapping takes place.

It was reported by Gers et al. (2000) that a SRN only memorized sequences of length between 3-5 units while a LSTM could memorize sequences of length longer than 1000 units. In this section, we study the memory of the SRN, LSTM and GRU.

### 2.1 MEMORY OF SRN

Here, for the ease of analysis, we use Elman's SRN model (Elman, 1990) with linear hidden state activation function and non-linear output activation function since such cell model is mathematically tractable and performance-wise equivalent to Jordan (1997) and Tensorflow's variations.

The SRN is described by the following two equations:

$$c_t = W_c c_{t-1} + W_{in} X_t, \tag{1}$$
$$h_t = f(c_t), \tag{2}$$

where subscript $t$ is the index of the time unit, $W_c \in \mathbb{R}^{N \times N}$ is the weight matrix for hidden state vector $c_{t-1} \in \mathbb{R}^N$, $W_{in} \in \mathbb{R}^{N \times M}$ is the weight matrix of input vector $X_t \in \mathbb{R}^M$, $h_t \in \mathbb{R}^N$ and $f(\cdot)$ is an element-wise non-linear function. Usually, $f(\cdot)$ is a hyperbolic-tangent or a sigmoid function. Throughout this paper, we omit the bias terms by putting them inside the corresponding weight matrices.

By induction, $c_t$ can be rewritten as

$$c_t = W_c^t c_0 + \sum_{k=1}^{t} W_c^{t-k} W_{in} X_k, \tag{3}$$

where $c_0$ is the initial internal state of the SRN. Typically, we set $c_0 = \underline{0}$. Then, Eq. (3) becomes

$$c_t = \sum_{k=1}^{t} W_c^{t-k} W_{in} X_k. \tag{4}$$

Let $\lambda_{\max}$ be the largest singular value of $W_c$. Then, we have

$$||W_c^{t-k} X_k|| \leq ||W_c||^{|t-k|} ||X_k|| = \lambda_{\max}^{|t-k|} ||X_k||. \tag{5}$$

Here, we are only interested in the case of memory decay when $\lambda_{\max} < 1$. Hence, the contribution of $X_k$, $k < t$, to $h_t$ decays at least in form of $\lambda_{\max}^{|t-k|}$. We conclude that SRN's memory decays at least exponentially with its memory length $|t - k|$.

## 2.2 MEMORY OF LSTM

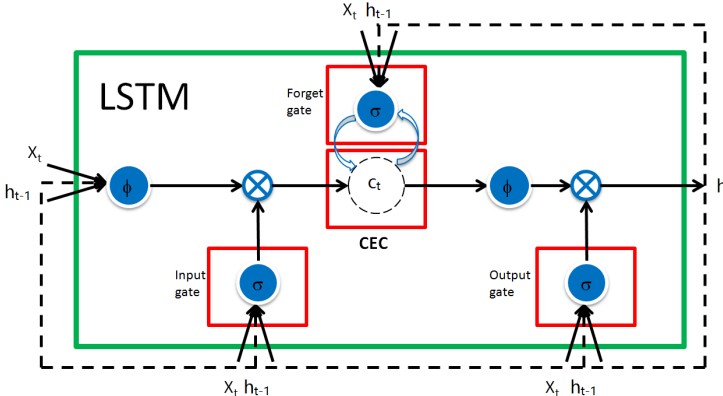

Figure 1: The diagram of a LSTM cell.

By following the work of Hochreiter & Schmidhuber (1997), we plot the diagram of a LSTM cell in Fig. 1. In this figure, $\phi$, $\sigma$ and $\otimes$ denote the hyperbolic tangent function, the sigmoid function and the multiplication operation, respectively. All of them operate in an element-wise fashion. The LSTM has an input gate, an output gate, a forget gate and a constant error carousal (CEC) module. Mathematically, the LSTM cell can be written as

$$c_t = \sigma(W_f I_t)c_{t-1} + \sigma(W_i I_t)\phi(W_{in} I_t), \tag{6}$$
$$h_t = \sigma(W_o I_t)\phi(c_t), \tag{7}$$

where $c_t \in \mathbb{R}^N$, column vector $I_t \in \mathbb{R}^{(M+N)}$ is a concatenation of the current input, $X_t \in \mathbb{R}^M$, and the previous output, $h_{t-1} \in \mathbb{R}^N$ (i.e., $I_t^T = [X_t^T, h_{t-1}^T]$). Furthermore, $W_f$, $W_i$, $W_o$ and $W_{in}$ are weight matrices for the forget gate, the input gate, the output gate and the input, respectively.

Under the assumption $c_0 = \underline{0}$, the hidden state vector of the LSTM can be derived by induction as

$$c_t = \sum_{k=1}^{t} \underbrace{\left[ \prod_{j=k+1}^{t} \sigma(W_f I_j) \right]}_{\text{forget gate}} \sigma(W_i I_k)\phi(W_{in} I_k), \tag{8}$$

By setting $f(\cdot)$ in Eq. (2) to a hyperbolic-tangent function, we can compare outputs of the SRN and the LSTM below:

$$\text{SRN:} \quad h_t^{SRN} = \phi\left( \sum_{k=1}^{t} W_c^{t-k} W_{in} X_k \right), \tag{9}$$

$$\text{LSTM:} \quad h_t^{LSTM} = \sigma(W_o I_t)\phi\left( \sum_{k=1}^{t} \underbrace{\left[ \prod_{j=k+1}^{t} \sigma(W_f I_j) \right]}_{\text{forget gate}} \sigma(W_i I_k)\phi(W_{in} I_k) \right). \tag{10}$$

We see from the above that $W_c^{t-k}$ and $\prod_{j=k+1}^{t} \sigma(W_f I_j)$ play the same memory role for the SRN and the LSTM, respectively.

If $W_f$ in Eq. (10) is selected such that

$$\min |\sigma(W_f I_j)| \geq \lambda_{\max}, \quad \forall \lambda_{\max} \in [0, 1),$$

then

$$\left| \prod_{j=k+1}^{t} \sigma(W_f I_j) \right| \geq \lambda_{\max}^{|t-k|}. \tag{11}$$

As given in Eqs. (5) and (11), the impact of input $I_k$ on output $h_t$ in the LSTM lasts longer than that of input $X_k$ in the SRN. This is the case if an appropriate weight matrix, $W_f$, of the forget gate is selected.

## 2.3 MEMORY OF GRU

The GRU was originally proposed for neural machine translation (Cho et al., 2014). It provides an effective alternative for the LSTM. Its operations can be expressed by the following four equations:

$$z_t = \sigma(W_z X_t + U_z h_{t-1}), \tag{12}$$
$$r_t = \sigma(W_r X_t + U_r h_{t-1}), \tag{13}$$
$$\tilde{h}_t = \phi(W X_t + U(r_t \otimes h_{t-1})), \tag{14}$$
$$h_t = z_t h_{t-1} + (1 - z_t)\tilde{h}_t, \tag{15}$$

where $X_t$, $h_t$, $z_t$ and $r_t$ denote the input vector, the hidden state vector, the update gate vector and the reset gate vector, respectively, and $W_z$, $W_r$, $W$, are trainable weight matrices. Its hidden state is also its output, which is given in Eq. (15). If we simplify the GRU by setting $U_z$, $U_r$ and $U$ to zero matrices, then we can obtain the following simplified GRU system:

$$z_t = \sigma(W_z X_t), \tag{16}$$
$$\tilde{h}_t = \phi(W X_t), \tag{17}$$
$$h_t = z_t h_{t-1} + (1 - z_t)\tilde{h}_t. \tag{18}$$

For the simplified GRU with the initial rest condition, we can derive the following by induction:

$$h_t = \sum_{k=1}^{t} \left[ \underbrace{\prod_{j=k+1}^{t} \sigma(W_z X_j)}_{\text{update gate}} \right] (1 - \sigma(W_z X_k))\phi(W X_k). \tag{19}$$

By comparing Eqs. (8) and (19), we see that the update gate of the simplified GRU and the forget gate of the LSTM play the same role. One can control the memory decay behavior of the GRU by choosing the weight matrix, $W_z$, of the update gate carefully.

## 3 EXTENDED-LONG SHORT-TERM MEMORY (ELSTM)

As discussed above, the LSTM and the GRU have longer memory by introducing the forget and the update gates, respectively. However, from Eq. 10 and Eq. 11, it can be seen that the impact of proceeding element to the current output at time step $t$ still fades quickly due to the presence of forget gate and update gate. And as we will show in the ELSTM design, this does not have to be the case.

In this section, we attempt to design extended-long short-term memory (ELSTM) cells and propose two new cell models:

- ELSTM-I: the extended-long short-term memory (ELSTM) with trainable input weight vector $s_i \in \mathbb{R}^N$, $i = 1, \cdots, t - 1$, where weights $s_i$ and $s_j$ (with $i \neq j$) are independent.

- ELSTM-II: the ELSTM-I with no forget gate.

These two cells are depicted in Figs. 2 (a) and (b), respectively.

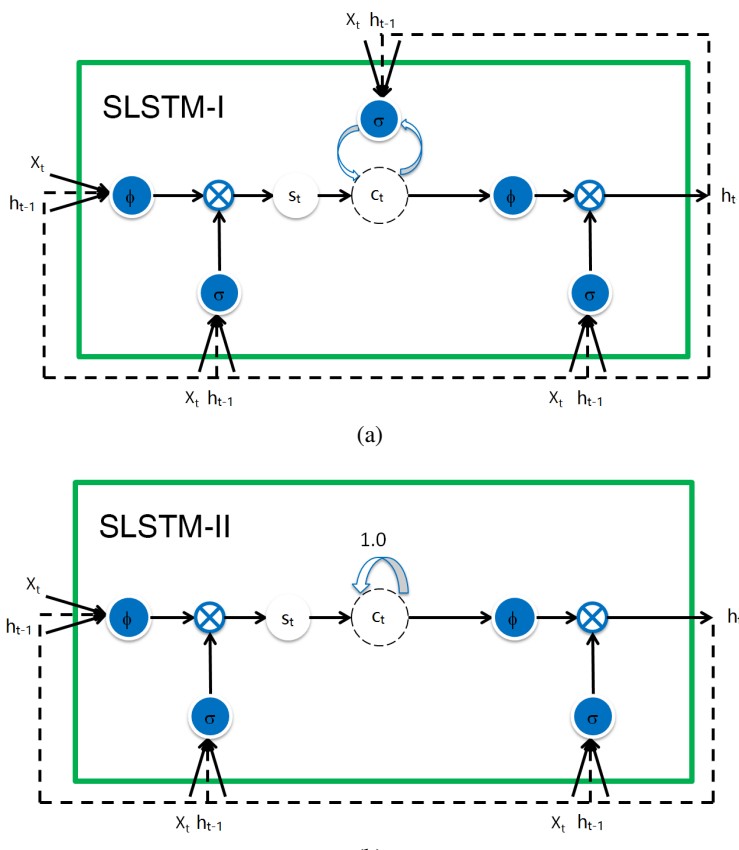

Figure 2: The diagrams of (a) the ELSTM-I cell and (b) the ELSTM-II cell.

The ELSTM-I cell can be described by

$$c_t = \sigma(W_f I_t)c_{t-1} + s_t\sigma(W_i I_t)\phi(W_{in} I_t), \tag{20}$$
$$h_t = \sigma(W_o I_t)\phi(c_t + b). \tag{21}$$

where $b \in \mathbb{R}^N$ is a trainable bias vector. The ELSTM-II cell can be written as

$$c_t = c_{t-1} + s_t\sigma(W_i I_t)\phi(W_{in} I_t), \tag{22}$$
$$h_t = \sigma(W_o I_t)\phi(c_t + b). \tag{23}$$

As shown above, we introduce scaling factor, $s_i$, $i = 1, \cdots, t-1$, to the ELSTM-I and the ELSTM-II to increase or decrease the impact of input $I_i$ in the sequence.

To prove that the proposed ELSTM-I has longer memory than LSTM, we first derive the closed form expression of $h_t$, which is:

$$h_t = \sigma(W_o I_t)\phi\left(\sum_{k=1}^{t} s_k\left[\prod_{j=k+1}^{t}\sigma(W_f I_j)\right]\sigma(W_i I_k)\phi(W_{in} I_k) + b\right). \tag{24}$$

We then pick $s_k$ such that:

$$\left|s_k\prod_{j=k+1}^{t}\sigma(W_f I_j)\right| \geq \left|\prod_{j=k+1}^{t}\sigma(W_f I_j)\right| \tag{25}$$

Compare Eq. 25 with Eq. 11, we conclude that ELSTM-I has longer memory than LSTM. As a matter of fact, $s_k$ plays a similarly role as the attention score in various attention models such

as Vinyals et al. (2015). The impact of proceeding elements to the current output can be adjusted (either increase or decrease) by $s_k$. The memory capability of ELSTM-II can be proven in a similarly fashion, so even ELSTM-II does not have forget gate, it is capable in attending to or forgetting a particular position of a sequence as ELSTM-I through the scaling factor.

The major difference between the ELSTM-I and the ELSTM-II is that fewer parameters are used in the ELSTM-II than those in the ELSTM-I. The numbers of parameters used by different RNN cells are compared in Table 1, where $X_t \in \mathbb{R}^M$, $h_t \in \mathbb{R}^N$ and $t = 1, \cdots, T$.

Table 1: Comparison of Parameter Numbers.

| Cell | Number of Parameters |
|------|----------------------|
| LSTM | $4N(M + N + 1)$ |
| GRU | $3N(M + N + 1)$ |
| ELSTM-I | $4N(M + N + 1) + N(T + 1)$ |
| ELSTM-II | $3N(M + N + 1) + N(T + 1)$ |

Although the number of parameters of ELSTM depends on the maximum length of a sequence in practice, the memory overhead required is limited. ELSTM-II requires less number of parameters than LSTM for typical lengthed sequence. From Table. 1, to double the number of parameters as compare to an ordinary LSTM, the length of a sentence needs to be 4 times the size of the word embedding size and number of cells put together. That is, in the case of Sutskever et al. (2014) with 1000 word embedding and 1000 cells, the sentence length needs to be $4 \times (1000 + 1000) = 8000$! In practice, most NLP problems whose input involves sentences, the length will be typically less than 100. In our experiment, sequence to sequence with attention (Vinyals et al., 2015) for maximum sentence length 100 (other model settings please refer to Table 2), ELSTM-I parameters uses 75M of memory, ELSTM-II uses 69.1M, LSTM uses 71.5M, and GRU uses 65.9M. Through GPU parallelization, the computational time for all four cells are almost identical with 0.4 seconds per step time on a GeForce GTX TITAN X GPU.

## 4 PROPOSED DEPENDENT BRNN (DBRNN) MODEL

We investigate the macro RNN model and propose a multitask model called dependent BRNN (DBRNN) in this section. The model is tasked to predict a output sequence $\{Y_t\}_{t=1}^{T'}$ ($Y_i \in \mathbb{R}^N$) based on the input sequence $\{X_t\}_{t=1}^{T}$ ($X_i \in \mathbb{R}^M$), where $T$ and $T'$ are the length of the input and output sequence respectively. Our proposal is inspired by the pros and cons of two RNN models – the bidirectional RNN (BRNN) model (Schuster & Paliwal, 1997) and the encoder-decoder model (Cho et al., 2014). In the following, we will first examine the BRNN and the encoder-decoder in Sec. 4.1 and, then, propose the DBRNN in Sec. 4.2.

### 4.1 BRNN AND ENCODER-DECODER

BRNN is modeling the conditional probability density function: $P(Y_t|\{X_i\}_{i=1}^{T})$. This output is a combination of the output of a forward and a backward RNN. Due to this bidirectional design, the BRNN can fully utilize the information of the entire input sequence to predict each individual output element. On the other hand, the BRNN does not utilize the predicted output in predicting $Y_t$. This makes elements in the predicted sequence $\hat{Y}_t = \text{argmax}_{Y_t} P(Y_t|\{X_i\}_{t=1}^{T})$ independent of each other.

This problem can be handled using the encoder-decoder model in form of $\hat{Y}_t = \text{argmax}_{Y_t} P(Y_t|\{\hat{Y}_i\}_{i=1}^{t-1}, \{X_i\}_{i=1}^{T})$. However, the encoder-decoder model is vulnerable to previous erroneous predictions in the forward path. Recently, the BRNN has been introduced in the encoder by Bahdanau et al. (2015), yet this design still does not address the erroneous prediction problem.

## 4.2 DBRNN Model and Training

Being motivated by observations in Sec. 4.1, we propose a multitask RNN model called DBRNN to fulfill the following objectives:

$$p_t = W^f p_t^f + W^b p_t^b \tag{26}$$

$$\hat{Y}_t = \underset{Y_t}{\operatorname{argmax}}\, p_t \tag{27}$$

$$\hat{Y}_t^f = \underset{Y_t}{\operatorname{argmax}}\, p_t^f, \tag{28}$$

$$\hat{Y}_t^b = \underset{Y_t}{\operatorname{argmax}}\, p_t^b, \tag{29}$$

where $W^f$ and $W^b$ are trainable weights. $p_t = P(Y_t|\{X_i\}_{i=1}^T)$, $p_t^f = P(Y_t|\{X_i\}_{i=1}^T, \{\hat{Y}_i^f\}_{i=1}^t)$ and $p_t^b = P(Y_t|\{X_i\}_{i=1}^T, \{\hat{Y}_i^b\}_{i=t}^{T'})$. The DBRNN has three learning objectives: the target sequence for the forward RNN prediction, the reversed target sequence for the backward RNN prediction, and, finally, the target sequence for the bidirectional prediction.

The DBRNN model is shown in Fig. 3. It consists of a lower and an upper BRNN branches. At each time step, the input to the forward and the backward parts of the upper BRNN is the concatenated forward and backward outputs from the lower BRNN branch. The final bidirectional prediction is the pooling of both the forward and backward predictions. We will show later that this design will make DBRNN robust to previous erroneous predictions.

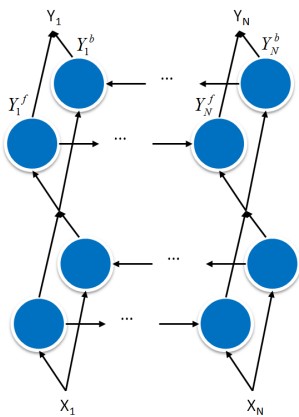

Figure 3: The DBRNN model.

Let $F(\cdot)$ be the cell function. The input is fed into the forward and backward RNN of the lower BRNN branch as

$$h_t^f = F_l^f\left(x_t, c_{l(t-1)}^f\right), \quad h_t^b = F_l^b\left(x_t, c_{l(t+1)}^b\right), \quad h_t = \begin{bmatrix} h_t^f \\ h_t^b \end{bmatrix}, \tag{30}$$

where $c$ denotes the cell hidden state and $l$ denotes the lower BRNN. The final output, $h_t$, of the lower BRNN is the concatenation of the output, $h_t^f$, of the forward RNN and the output, $h_t^b$, of the backward RNN. Similarly, the upper BRNN generates the final output $p_t$ as

$$p_t^f = F_u^f\left(h_t, c_{u(t-1)}^f\right), \quad p_t^b = F_u^b\left(h_t, c_{u(t+1)}^b\right), \quad p_t = W^f p_t^f + W^b p_t^b, \tag{31}$$

where $u$ denotes the upper BRNN. To generate forward prediction $\hat{Y}_t^f$ and backward prediction $\hat{Y}_t^b$, the forward and backward paths of the upper BRNN branches are trained separately with the target sequence and the reversed target sequence, respectively. The results of the upper forward and backward RNN are then combined to generate the final result.

There are three errors: prediction error of $\hat{Y}_t^f$ denoted by $e_f$, prediction error of $\hat{Y}_t^b$ denoted by $e_b$ and prediction error of $\hat{Y}_t$ denoted by $e$. To train this network, $e_f$ is back propagated through time

to the upper forward RNN and the lower BRNN, $e_b$ is back propagated through time to the upper backward RNN and the lower BRNN, and $e$ is back propagated through time to the entire model.

To show that DBRNN is more robust to previous erroneous predictions than one-directional models, we compare the cross entropy of them as follows:

$$l = -\sum_{k=1}^{K} p_{tk}\log(\hat{p}_{tk}), \tag{32}$$

where $K$ is the total number of classes (e.g. the size of vocabulary for language tasks). $p_t$ is the ground truth distribution which is an one- hot vector such that: $p_{tk} = \mathbb{I}(p_{tk} = k'), \forall k \in 1, ..., K$, where $\mathbb{I}$ is the indicator function, $k'$ is the ground truth label of the $t$th output. $\hat{p}_t$ is the predicted distribution.

From Eq. 26, $l$ can be further expressed as:

$$l = -\sum_{k=1}^{K} p_{tk}\log(W_k^f \hat{p}_{tk}^f + W_k^b \hat{p}_{tk}^b), \tag{33}$$

$$= -\log(W_{k'}^f \hat{p}_{tk'}^f + W_{k'}^b \hat{p}_{tk'}^b), \tag{34}$$

We can pick $W_{tk'}^f$ and $W_{tk'}^b$ such that $W_{k'}^f \hat{p}_{tk'}^f + W_{k'}^b \hat{p}_{tk'}^b$ is greater than $\hat{p}_{tk'}^f$ and $\hat{p}_{tk'}^b$. Which leads to $l < -\sum_{k=1}^{K} \log(\hat{p}_{tk}^f)$ and $l < -\sum_{k=1}^{K} \log(\hat{p}_{tk}^b)$, which means DBRNN can have less cross entropy than one-directional prediction by pooling expert opinions from that two predictions.

It is worthwhile to compare the DBRNN and the solution in Cheng et al. (2016). Both of them have a bidirectional design for the output. However, there exist three main differences. First, the DBRNN is a general design for the sequence-in-sequence-out (SISO) problem without being restricted to dependency parsing. The target sequences in training $\hat{Y}_t^f$, $\hat{Y}_t^b$ and $\hat{Y}_t$ are the same for the DBRNN. In contrast, the solution in Cheng et al. (2016) has different target sequences. Second, the attention mechanism is used by Cheng et al. (2016) but not in the DBRNN. Third, The encoder-decoder design is adopted by in Cheng et al. (2016) but not in the DBRNN.

## 5 EXPERIMENTS

### 5.1 EXPERIMENTAL SETUP

We conduct experiments on two problems: part of speech (POS) tagging and dependency parsing (DP). The POS tagging task is an easy one which requires shorter memory while the DP task needs much longer memory and has more complex relations between the input and the output.

In the experiments, we compare the performance of five RNN models under two scenarios: 1) $I_t = X_t$, and 2) $I_t^T = [X_t^T, h_{t-1}^T]$. The five RNN models are the basic one-directional RNN (basic RNN), the BRNN, the sequence-to-sequence (a variation of encoder-decoder) RNN, sequence-to-sequence with attention and the DBRNN with four cell designs (LSTM, GRU, ELSTM-I and ELSTM-II). When $I_t = X_t$, we do not include the GRU cell since it inherently demands $I_t^T = [X_t^T, h_{t-1}^T]$. For the DBRNN, we show the results for $Y_t$ (denoted by "DBRNN combined") and $Y_t^f$ (denoted by "DBRNN forward"), which is the prediction from the forward path of the upper BRNN branch. We do not include the result for $Y_t^b$, which is the prediction from the backward path of the upper BRNN branch since the performance of the backward RNN path of the upper BRNN branch is poorer.

The training dataset used for both problems are from the Universal Dependency 2.0 English branch (UD-English). It contains 12543 sentences and 14985 unique tokens. The test dataset for both experiments is from the test English branch (gold, en.conllu) of CoNLL 2017 shared task development and test data.

In the experiment, the lengths of the input and the target sequences are fixed. Sequences longer than the maximum length will be truncated. If the sequence is shorter than the fixed length, a special pad

symbol will be used to pad the sequence. Similar technique called bucketing is also used for some popular models such as Sutskever et al. (2014). The input to the POS tagging and the DP problems are the stemmed and lemmatized sequences (column 3 in CoNLL-U format). The target sequence for POS tagging is the universal POS tag (column 4). The target sequence for DP is the interleaved dependency relation to the headword (relation, column 8) and its position (column 7). As a result, the length of the actual target sequence (rather than the preprocessed fixed-length sequence) for DP is twice of the length of the actual input sequence.

Table 2: Network and training details

| | |
|---|---|
| Input/output sequence fixed length | 100/100 |
| Number of RNN layers | 1 |
| Embedding layer vector size | 512 |
| Number of RNN cells | 512 |
| Batch size | 20 |
| Training steps | 140000 |
| Learning rate | 0.5 |
| Training optimizer | AdaGrad(Duchi, 2011) |
| Maximum gradient norm | 5 |

The input is first fed into a trainable embedding layer (Bengio et al., 2003) before it is sent to the actual network. Table 2 shows the detailed network and training specifications. It is important to point out that we do not finetune network parameters or apply any engineering trick for the best possible performance since our main goal is to compare the performance of the LSTM, GRU, ELSTM-I and ELSTM-II four cells under various macro-models.

## 5.2 EXPERIMENTAL RESULTS

The results of the POS tagging problem with $I_t = X_t$ and $I_t^T = [X_t^T, h_{t-1}^T]$ are shown in Tables 3 and 4, respectively. Among all possible combinations, the DBRNN with the LSTM cell has the highest accuracy (89.16%) for $I_t = X_t$ while the DBRNN with the GRU cell has the (89.74%) has the highest accuracy for $I_t^T = [X_t^T, h_{t-1}^T]$.

Table 3: POS tagging test accuracy, $I_t = X_t$ (%)

| | LSTM | ELSTM-I | ELSTM-II |
|---|---|---|---|
| BASIC RNN | **85.38** | 85.30 | 84.35 |
| BRNN (Schuster & Paliwal, 1997) | **88.49** | 82.84 | 79.14 |
| Seq2seq (Sutskever et al., 2014) | 25.83 | 24.87 | **31.43** |
| Seq2seq with Attention (Vinyals et al., 2015) | 27.97 | **78.98** | 42.05 |
| DBRNN Combined | **89.16** | 83.69 | 81.08 |
| DBRNN Forward | **88.93** | 83.54 | 81.08 |

Table 4: POS tagging test accuracy, $I_t^T = [X_t^T, h_{t-1}^T]$ (%)

| | LSTM | GRU | ELSTM-I | ELSTM-II |
|---|---|---|---|---|
| BASIC RNN | 86.98 | **87.09** | 85.57 | 85.56 |
| BRNN | 88.94 | **89.26** | 83.48 | 82.57 |
| Seq2seq | 24.73 | 33.79 | 34.09 | **52.96** |
| Seq2seq with Attention | 34.10 | 73.65 | **80.90** | 54.53 |
| DBRNN Combined | 89.67 | **89.74** | 84.25 | 84.41 |
| DBRNN Forward | 89.46 | **89.53** | 84.05 | 84.44 |

The results of the DP problem with $I_t = X_t$ and $I_t^T = [X_t^T, h_{t-1}^T]$ are shown in Tables 5 and 6, respectively. The ELSTM-I and ELSTM-II cells perform better than the LSTM and the GRU cells. Among all possible combinations, the sequence-to-sequence with attention combined with ELSTM-I has the best performance. It has an accuracy of 60.19% and 66.72% for the former and the latter,

respectively. Also, the basic RNN often outperforms BRNN for the DP problem as shown in Tables 5 and 6. This can be explained by that the basic RNN can access the entire input sequence when predicting the latter half of the output sequence since the target sequence is twice as long as the input. The other reason is that the BRNN can easily overfit when predicting the headword position.

Table 5: DP test accuracy, $I_t = X_t$ (%)

|  | LSTM | ELSTM-I | ELSTM-II |
|---|---|---|---|
| BASIC RNN | 15.14 | 38.36 | **42.52** |
| BRNN | 14.74 | **39.24** | 35.78 |
| Seq2seq | 24.37 | 30.04 | **35.67** |
| Seq2seq with Attention | 21.56 | **60.19** | 45.15 |
| DBRNN Combined | 25.26 | **54.39** | 53.80 |
| DBRNN Forward | 25.71 | **53.67** | 52.61 |

Table 6: DP test accuracy, $I_t^T = [X_t^T, h_{t-1}^T]$ (%)

|  | LSTM | GRU | ELSTM-I | ELSTM-II |
|---|---|---|---|---|
| BASIC RNN | 44.12 | 47.49 | 54.52 | **56.02** |
| BRNN | 32.46 | 27.83 | **54.14** | 47.72 |
| Seq2seq | 27.67 | 29.94 | 40.85 | **48.73** |
| Seq2seq with Attention | 31.47 | 53.70 | **66.72** | 51.24 |
| DBRNN Combined | 56.89 | 51.32 | **60.30** | 58.28 |
| DBRNN Forward | 58.30 | 53.17 | **59.81** | 58.05 |
| Bi-attention (Cheng et al., 2016) [1] | | 61.29 | | |

We see from Tables 3 - 6 that the two DBRNN models outperform both BRNN and sequence-to-sequence (without attention) in both POS tagging and DP problems regardless of used cells. This shows the superiority of introducing the expert opinion pooling from both the input and the predicted output.

Furthermore, the proposed ELSTM-I and ELSTM-II outperform the LSTM and the GRU by a significant margin for complex language tasks. This demonstrates that the scaling factor in the ELSTM-I and the ELSTM-II does help the network retain longer memory with better attention. ELSTMs even outperform Cheng et al. (2016), which is designed specifically for DP. For the POS tagging problem, the ELSTM-I and the ELSTM-II do not perform as well as the GRU or the LSTM. This is probably due to the shorter memory requirement of this simple task. The ELSTM cells are over-parameterized and, as a result, they converge slower and tend to overfit the training data.

The ELSTM-I and the ELSTM-II perform particularly well for sequence-to-sequence (with and without attention) model. The hidden state $c_t$ of the ELSTMs is more expressive in representing patterns over a longer distance. Since the sequence-to-sequence design relies on the expressive power of the hidden state, the ELSTMs do have an advantage.

We compare the convergence behavior of $I_t = X_t$ and $I_t^T = [X_t^T, h_{t-1}^T]$ with the LSTM, the ELSTM-I and the ELSTM-II cells for the DP problem in Fig. 4. We see that the ELSTM-I and the ELSTM-II do not behave very differently between $I_t = X_t$ and $I_t^T = [X_t^T, h_{t-1}^T]$ as the LSTM does. This shows the effectiveness of the ELSTM-I and the ELSTM-II design regardless of the input. More performance comparison will be provided in the Appendix.

## 6 CONCLUSION AND FUTURE WORK

The memory decay behavior of the LSTM and the GRU was investigated and explained by mathematical analysis. Although the memory of the LSTM and the GRU fades slower than that of the SRN, it may not be long enough for complicated language tasks such as dependency parsing. To

---

[1]The result is generated by using exactly the same settings in Table. 2. We do not feed in the network with information other than input sequence itself.

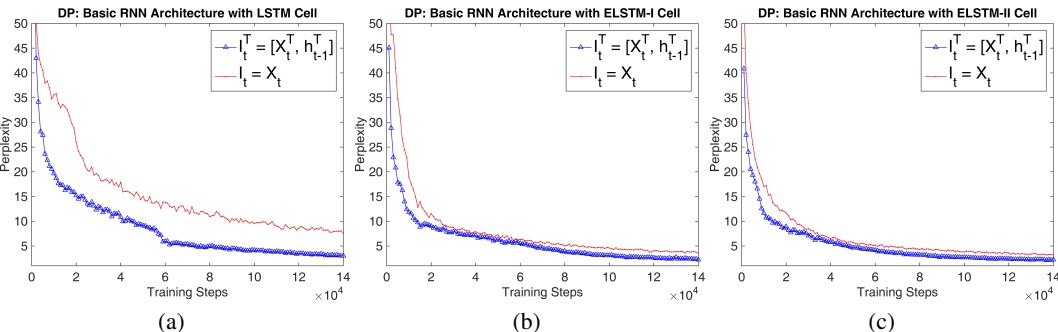

Figure 4: Training perplexity of the basic RNN with $I_t = X_t$ and $I_t^T = [X_t^T, h_{t-1}^T]$ for the DP problem.

enhance the memory length, two cells called the ELSTM-I and the ELSTM-II were proposed. Furthermore, we introduced a new RNN model called the DBRNN that has the merits of both the BRNN and the encoder-decoder. It was shown by experimental results that the ELSTM-I and ELSTM-II outperforms other designs by a significant margin for complex language tasks. The DBRNN design is superior to BRNN as well as sequence-to-sequence models for both simple and complex language tasks. There are interesting issues to be further explored. For example, is the ELSTM cell also helpful in more sophisticated RNN models such as the deep RNN? Is it possible to make the DBRNN deeper and better? They are left for future study.

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

APPENDIX: MORE EXPERIMENTAL RESULTS

In the appendix, we provide more experimental results to shed light on the convergence performance in the training of various models with different cells for the POS tagging and the DP tasks. First, we compare the training perplexity between $I_t = X_t$ and $I_t^T = [X_t^T, h_{t-1}^T]$ for various models with the LSTM, the ELSTM-I and the ELSTM-II cells in Figs. 5 - 8. Then, we examine the training perplexity with $I_t^T = [X_t^T, h_{t-1}^T]$ for various models with different cells in Figs. 9 - 12.

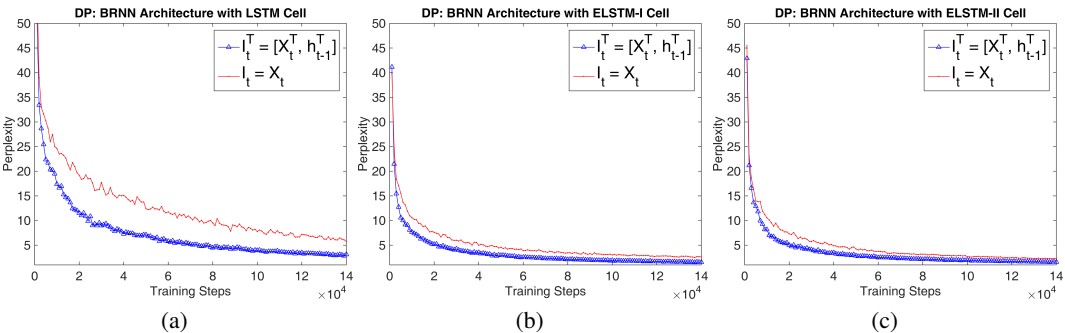

Figure 5: The training perplexity of the BRNN model with $I_t = X_t$ and $I_t^T = [X_t^T, h_{t-1}^T]$ for the DP task.

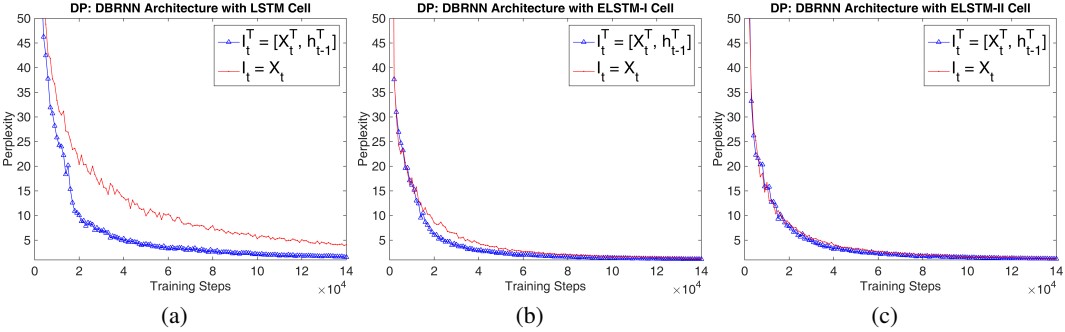

Figure 6: The training perplexity of the DBRNN model with $I_t = X_t$ and $I_t^T = [X_t^T, h_{t-1}^T]$ for the DP task.

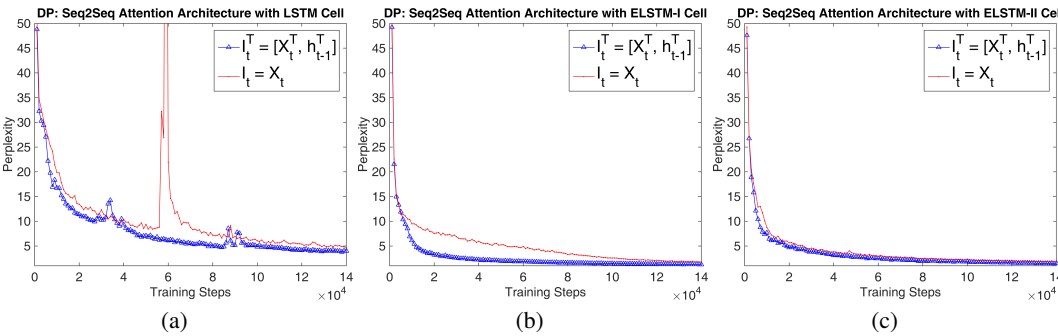

Figure 7: The training perplexity of the sequence-to-sequence model with $I_t = X_t$ and $I_t^T = [X_t^T, h_{t-1}^T]$ for the DP task.

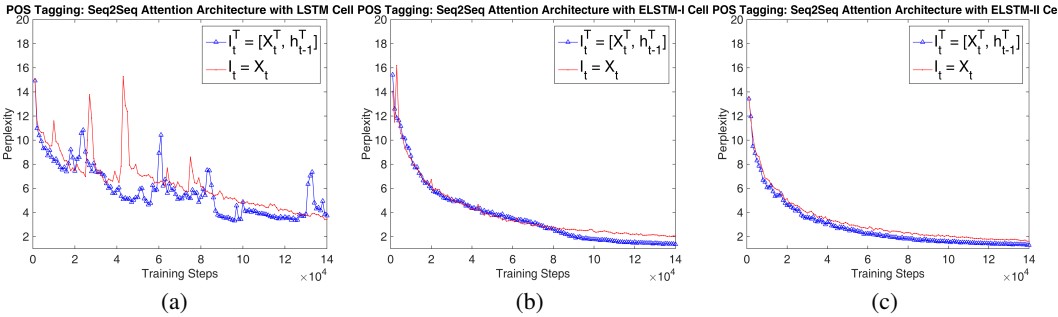

Figure 8: The training perplexity of the sequence-to-sequence model with $I_t = X_t$ and $I_t^T = [X_t^T, h_{t-1}^T]$ for the POS tagging task.

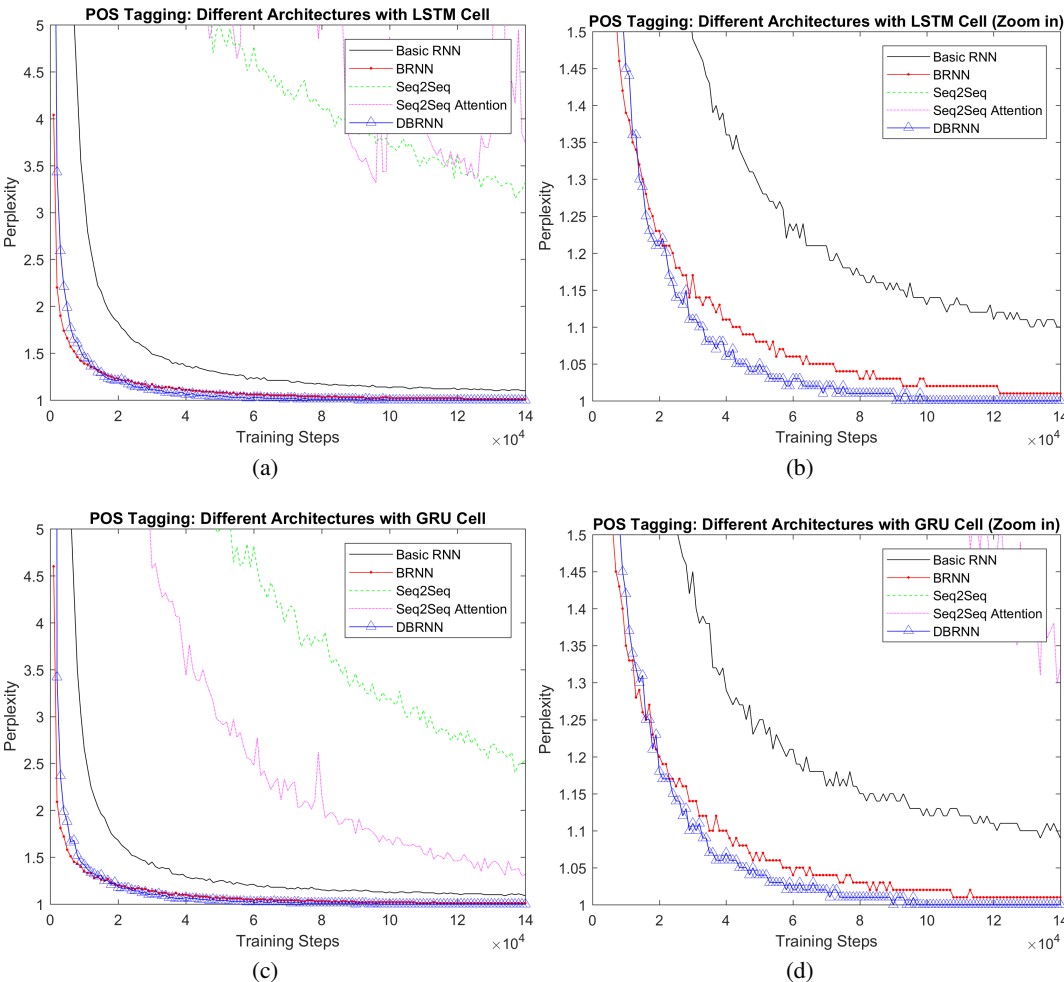

Figure 9: The training perplexity of different models with the LSTM (top left), its zoom-in (top right) and the GRU (bottom left), its zoom-in (bottom right) for the POS tagging.

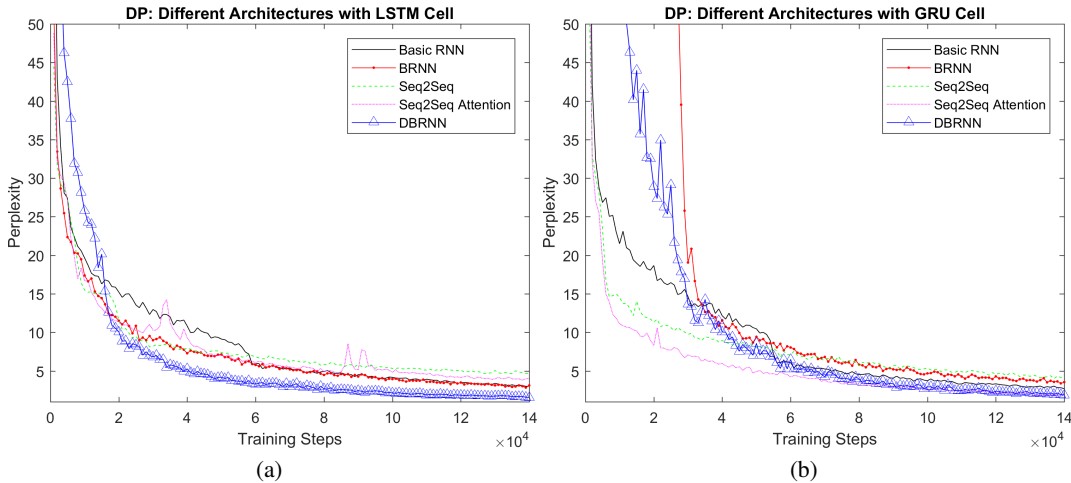

(a)         (b)

Figure 10: The training perplexity of different models with the LSTM (left) and the GRU (right) cells for the DP task.

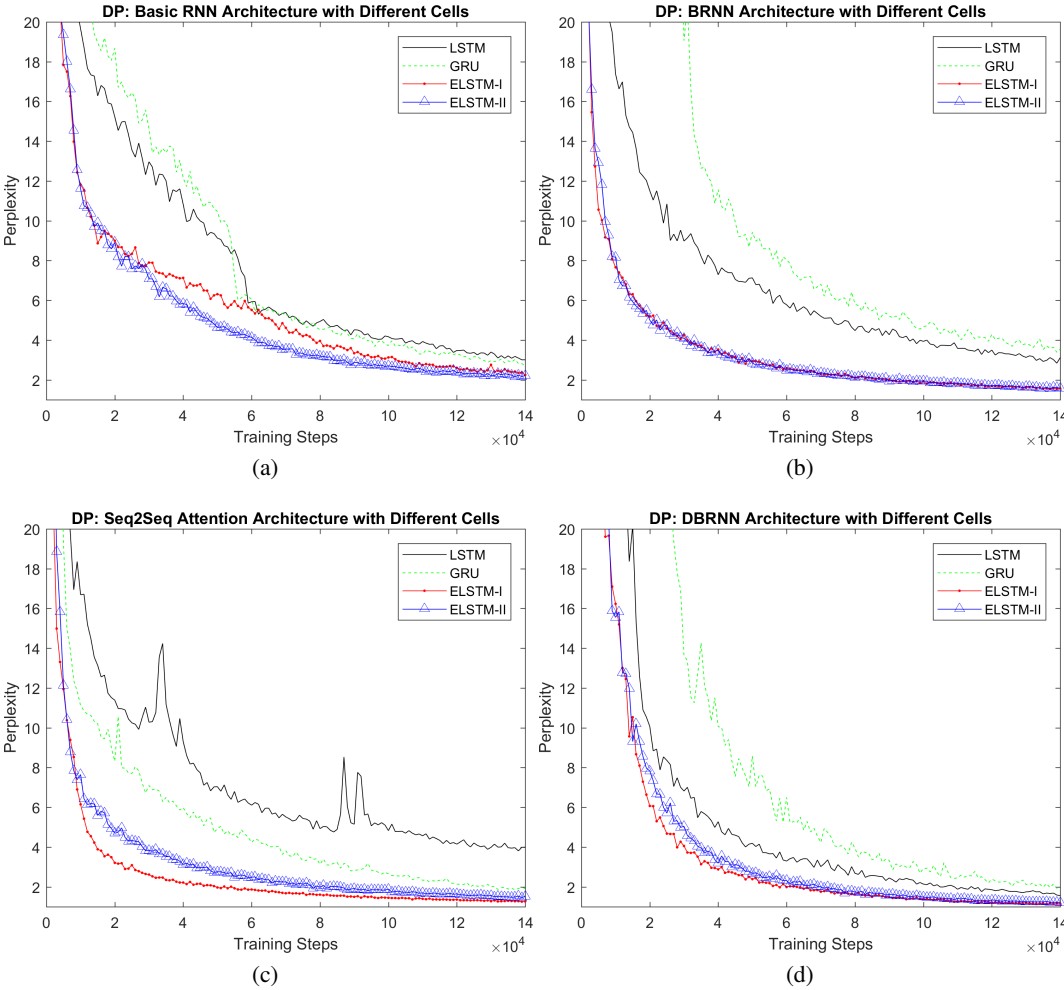

(a)         (b)

(c)         (d)

Figure 11: The training perplexity for the basic RNN (top left), the BRNN (top right), the sequence-to-sequence (bottom left) and the DBRNN models (bottom right) for the DP task.

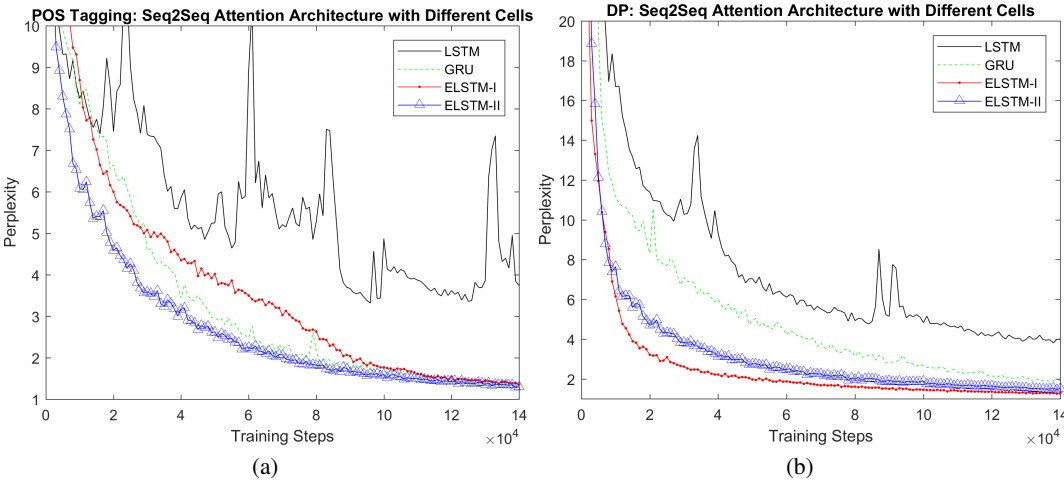

Figure 12: The training perplexity of the sequence-to-sequence model with the ELSTM-II cell for the POS tagging (left) and for the DP (right) tasks.

