# OpenReview forum: "Dependent Bidirectional RNN with Extended-long Short-term Memory"
_ICLR.cc/2018/Conference — Reject_

### Official Review · AnonReviewer1 · 2017-11-19
**serious presentation issues**

**Rating:** 3
**Confidence:** 4

**Review:**

The paper proposes a new recurrent cell and a new way to make predictions for sequence tagging. It starts with a theoretical analysis of memory capabilities in different RNN cells and goes on with experiments on POS tagging and dependency parsing. There are serious presentation issues in the paper, which make it hard to understand the ideas and claims.

First, I was not able to understand the message of the theoretical analysis from Section 2 and could not see how it is different from similar derivations (i.e. using a linearized version of an RNN and eigenvalue decomposition) that can be found in many other papers, including (Bengio et al, 1994) and (Pascanu et al, 2013). Novelty aside, the analysis has presentation issues. SRN is introduced without a nonlinearity from the beginning, although normally it should have one. From the classical upper bound with a power of the largest singular value the paper concludes that “Clearly, the memory will explode if \lambda_{max} > 1”, which is not true: the memory *may* explode, having an exponentially growing upper bound does not mean that it *will* explode. The notation chosen from LSTM is  different from the standard in deep learning community and was very hard to understand (Y_t is used instead of h_t, and h_t is used instead of c_t). This notation also does not seem consistent with the rest of the paper, for example Equations 28 and 29 suggest that Y_t are discrete outputs and not vectors.

The novel cell SLSTM-I is meant to be different from LSTM by addition of “input weight vector c_i”, but is not explained where c_i come from. Are they trainable vectors, one for each time step? If yes, then how could such a cell be applied to sequence which are longer than the training ones?

Equations 28, 29, 30 describe a very unusual kind of a Bidirectional Recurrent Network. To the best of my knowledge it is much more common to make one prediction based on future and past information, whereas the paper describes an approach in which first predictions are made separately based on the past and on the future. It is also very common to use several BiRNN layers, whereas the paper only uses one. As for the proposed DBRNN method, unfortunately, I was not able to understand it.

I also have concerns regarding the experiments. Why is seq2seq without attention is used? On such small datasets attention is likely to make a big difference. What’s the point of reporting results of an LSTM without output nonlinearity (Table 5)?

To sum up, the paper needs a lot work on many fronts, but most importantly, presentation should be improved.

---

> ### Author Response · Authors · 2017-12-09
> **Author Response 1(to be continued)**
>
> Dear Reviewer, we do apologize for the presentation style that confuses readers. We will definitely clarify the notations as you suggested, and some basic concepts to develop our ideas.
> We have also added new experiments as you suggested, as the results (which will be elaborated in the following feedback points) suggests that our design do have an edge as compare to existing RNN cell designs.
> Hopefully, after our feedback and coming revision, you can have better idea of what this work is about.
> We also feel that this paper introduces some noteworthy ideas that may benefit the NLP and RNN communities for their future research
>
> 1. How the mathematical analyzing part differs to other RNN works?
> Unlike many other RNN/NLP papers which focus on gradient vanishing/exploding problem which happens in training process, this work primarily focus on the "memory" behavior of RNN once it is trained. That answers your question of how this paper is different to (Bengio et al, 1994) and (Pascanu et al, 2013). Here, we analyze the memory of different RNN cells by looking into their output as a function of the input, and derive the relationship between current output to previous inputs.
>
> 2. SRN formulation is problematic
> We should have mentioned that the SRN model we adopted is from Elman,1990 with linear hidden state activation function and non-linear output activation function, so our SRN formulation is not a linearized version. There are some variations of SRN, including Elman,1990 and Jordan,1997. Tensorflow also has its own variation of SRN. The SRN models that feed the previous output back to the input is Jordan,1990 and Tensorflow. For the Elman's SRN model, the hidden state is fed back to the system instead.
> The reason we chose Elman's SRN model is that it is mathematically tractable, and it can be analyzed using induction. For the other two models, however, due to the presence of non-linear output activation f(), such analysis is impossible. In addition, all three SRN models are similar to each other performance-wise.
> So the choice of Elman's model, in our assessment, is valid. It opens the door for the ensuing mathematical comparison between LSTM and SRN, which, to our knowledge, the first time in literature.
>
> 3. Upper bound does not mandate memory explosion
> Yes, we agree with your conclusion here. In our revised paper, we will strictly limit our discussion to the case that the largest singular value is less than one, and then the memory of SRN will decay at least exponentially.
> Nevertheless, the derivation of SRN memory does not impact our major contribution: the relative memory capabilities between LSTM and SRN.
>
> 4. Notations
> Thank you for pointing this out, we will change the notations as you suggested.
>
> 5. Where the "c_i" comes from?
> Yes, it is a trainable vector, one for each time step.
> To use SLSTM, one needs to decide the maximum sequence length for implementation. Such practice is also used for some popular models like sequence to sequence (Ilya Sutskever,2014) and sequence to sequence with attention (O.Vinyals, 2015). Please refer to the tensorflow implementation for bucketing here:
> https://github.com/tensorflow/nmt
> Or here from line 1145:	   https://github.com/tensorflow/tensorflow/blob/master/tensorflow/contrib/legacy_seq2seq/python/ops/seq2seq.py
> In terms of whether SLSTM is efficient in number of parameters and computational complexity, here is our response to a similar question to reviewer 3:
> "However, the memory overhead required for SLSTM is limited. In addition, SLSTM-II requires less number of parameters than LSTM for typical lengthed sequence. Please take a look at table 1 on page 5: to double the number of parameters as compare to an ordinary LSTM, the length of a sentence needs to be 4 times the size of the word embedding size and number of cells put together. That is, in the case of Ilya Sutskever,2014 with 1000 word embedding and 1000 cells, the sentence length needs to be 4x(1000+1000) = 8000! In practice, most NLP problems whose input involves sentences, the length wil be typically less than 100. As a matter of fact, the tensorflow implementation of O.Vinyals, 2015 for machine translation, the maximum sentence length is truncated up to 50. In our experiment, sequence to sequence with attention (O.Vinyals, 2015) for maximum sentence length 100 (other model settings please refer to Table 2), SLSTM-I parameters uses 75M of memory, SLSTM-II uses 69.1M, LSTM uses 71.5M, and GRU uses 65.9M. Through GPU parallization, the computational time for all four cells is almost identical with 0.4 seconds per step time on a TITAN X 1080 GPU.
> So the bottom line is: unless you are dealing with sequence length of more than hundreds of thousands, you should not worry about the memory (or the waste of it) or computational complexity issue."

---

> > ### Author Response · Authors · 2017-12-09
> > **Author Response2**
> >
> > 6. What is DBRNN?
> > As summed up by reviewer3, DBRNN is a multi-task model with three learning objectives: the one-directional forward target sequence, the one-directional backward target sequence, and the bi-directional target sequence.
> > One can certainly make DBRNN deeper. In our experiments, we are more concerned with the fundamental model capabilities rather than how many RNN layers the model can have. As reported in (Ilya Sutskever,2014), the sequence-to-sequence model gives good performance for machine translation up to 4 layers. The optimal number of layers is itself an ongoing research topic (Razvan Pascanu, 2014). Such topic is beyond the scope of this work.
> > What makes DBRNN stand out to sequence-to-sequence model is it is mathematically robust to previous erroneous predictions, which is shown as below:
> > Denote p_t as the ground truth distribution of the output at time step t, \hat{p}_t is the final prediction from the model, \hat{p}^f_t is the forward prediction, \hat{p}^b_t is the backward prediction. And we have \hat{p}_t = W^f \hat{p}^f_t + W^b \hat{p}^b_t (Eq. 36 on page 7, we just change the notation from Y to p as suggested). The p_t is an one-hot vector in the form of p_t = \mathbbm{I}(p_{tk} = k'), \forall k \in 1,...,K. Where \mathbmm{I} is indicator function, k' is the ground truth label. Then the cross entropy of DBRNN is:
> > l = -\sum^K_{k=1} p_{tk} log( \hat{p}_{tk} )
> > where K is the total number of classes (size of vocabulary)
> > It can be further expressed as:
> > l = -\sum^K_{k=1} p_{tk} log( W^f_k \hat{p}^f_{tk} + W^b_k \hat{p}^b_{tk})
> >   = -log( W^f_{k'} \hat{p}^f_{tk'} + W^b_{k'} \hat{p}^b_{tk'})
> > We can pick W^f_{tk'} and W^b_{tk'} such that l < -\sum^K_{k=1} log(\hat{p}^f_{tk}) and l < -\sum^K_{k=1} log(\hat{p}^b_{tk}). Which means DBRNN can have less cross entropy than one-directional prediction by pooling expert opinions from that two predictions.
> > In the experiment results section, DBRNN with GRU gives the best performance for POS, it also outperforms BRNN and sequence-to-sequence in all language tasks which validates our claim.
> >
> > 7. Need to compare to sequence-to-sequence with attention
> > Thank you for this suggestion. We carried out experiments for sequence-to-sequence with attention, the results show that our design SLSTM outperforms other cells by around 10%
> > POS, I^T_t = [X^T_t, Y^T_{t-1}]:
> > 			                              LSTM	GRU		SLSTM-I		SLSTM-II
> > seq2seq with attention	    34.10%	73.65%	        80.90%		54.53%
> > DP, I^T_t = [X^T_t, Y^T_{t-1}]:
> > 			                              LSTM	GRU		SLSTM-I		SLSTM-II
> > seq2seq with attention	    31.47%	53.70%	        66.72%		51.24%
> > It is interesting to see that seq2seq with attention with SLSTM-I also outperform DBRNN. We will do our detailed analysis on these updated results. We also include the Cheng's result for DP, the result is:
> > DP
> > 			                 GRU
> > Cheng's bi-attention:	61.29%
> > As to why we also show the results when the input to the model does not include previous output (I_t = X_t), the point is to show that our proposal is robust to different inputs, as shown in Fig. 4. It also substantiates our analysis of SLSTM.

---

> ### Author Response · Authors · 2017-12-13
> **Updated Revision**
>
> Dear Reiviewer
>
> We have revised our paper according to your feedback, we have made some important revisions:
> 1. We have changed the cell name from super-long short-term memory (SLSTM) to extended-long short-term memory (ELSTM) as suggested by reviewer 2.
> 2. The mathematical proof of the memory capability of ELSTM and the robustness of DBRNN has been added
> 3. Argument of the added number of parameters of ELSTM does not affect its practicability has been added
> 4. Experiments for sequence-to-sequence with attention and Cheng's bi-attention model have been added, the results further confirm our cell and RNN model design.
>
> best

---

### Official Review · AnonReviewer2 · 2017-11-27
**Generally a nice approach, but the paper has several issues, especially, recent SotA should be taken into account**

**Rating:** 4
**Confidence:** 4

**Review:**


This paper introduces a different form of Memory cell for RNN which has more capabilities of long-term memorizing. Furthermore, it presents and efficient architecture for sequence-to-sequence mapping.

While the claim of the paper sounds very ambitious and good, the paper has several flaws. First of all, the mathematical analysis is a bit problematic. Of course, it is known that Simple Recurrent Networks (SRN) have a vanishing gradient problem. However, the way you proof it is not correct, as you ignore the application of f() for calculating the output (which is routed to the input) and you use an upper bound to show a general behaviour.
The analysis of the Memory capabilities of LSTM is a bit simplified, however, it is okay. Note, that various experiments by Schmidhuber's group, as well as Otte & al have shown that LSTM can generalize and memorize to sequences of more than a million time steps, if the learning rates is small enough.

The extended memory which the authors call SLSTM-I has similar memory capabilities as LSTM. The other one (SLSTM-II) looses the capability of forgetting as it seems. An analysis would be crucial in this paper to show the benefits mathematically.

The authors should have a look at "Evolving memory cell structures for sequence learning" by Justin Bayer, Daan Wierstra, Julian Togelius and J¨urgen Schmidhuber, published in 2009. Note that the SLSTM belongs to the family of networks which could be generated by that paper as well.

Also "Neural Architecture Search with Reinforcement Learning" by Barret Zoph and Quoc V. Le would be interesting.

In your experiments it would be fair to compare to Cheng et al. 2016

I suggest the authors being more modest with the name of the memory cell as well as with the abstract (especially in the POS experiment, SLSTM is not superior)

---

> ### Author Response · Authors · 2017-12-09
> **Author Response**
>
> Thank you for the valuable input for this paper, especially the literatures you point to reference.
> To address your concerns of this work, we would like to answer them as follows and reflect these answers to our next revision:
>
> 1. The formulation of SRN is incorrect
> Thank you for the careful examination. We should have mentioned that the SRN model we adopt is from Elman,1990 with linear hidden state activation function. There are some variations of SRN, including Elman,1990 and Jordan,1997. Tensorflow also has its own variation of SRN. The SRN models that feed the previous output back to the input is Jordan,1990 and Tensorflow. For the Elman's SRN model, the hidden state is fed back to the system instead.
> The reason we chose Elman's SRN model is that it is mathematically tractable, and it can be analyzed using induction. For the other two models, however, due to the presence of non-linear output activation f(), such analysis is impossible. In addition, all three SRN models are similar to each other performance-wise.
> So the choice of Elman's model, in our assessment, is valid. It opens the door for the ensuing mathematical comparison between LSTM and SRN, which, to our knowledge, the first time in literature.
>
> 2. Showing upper bound is inadequate
> It is important to point out that in most RNN literature, the point of interest is about gradient vanishing/exploding problem which happens in the training process. And on that front, Pascanu Razvan, 2013 already uses upper bound to show the necessary conditions for gradient vanishing and exploding.
> This work focus more on the memory decay problem once the RNN model is trained, and we follow similar proofing procedure. In our SRN analysis, the upper bound indicates once the largest singular value of the weight of hidden state is less than one, the memory of SRN will decay at least exponentially. Such conclusion is solid.
>
> 3. Show the benefits of SLSTM (we are considering to change the name to make it sound modest, will appear soon after our next revision) mathematically
> Thank you for reminding us the crucial missing part!
> Before we show how mathematically SLSTM has longer memory than LSTM, we need to first point out that in our experiment results section, SLSTM demonstrates qualitative difference as compared to LSTM for complex NLP problems.
>  As suggested by other two reviewers, we also carried experiments on sequence to sequence with attention (O.Vinyals, 2015), the results are as follows:
> POS, I^T_t = [X^T_t, Y^T_{t-1}]:
> 			                              LSTM	GRU		SLSTM-I		SLSTM-II
> seq2seq with attention	    34.10%	73.65%	        80.90%		54.53%
> DP, I^T_t = [X^T_t, Y^T_{t-1}]:
> 			                               LSTM	GRU		SLSTM-I		SLSTM-II
> seq2seq with attention	    31.47%	53.70%	        66.72%		51.24%
> It is interesting to see that seq2seq with attention with SLSTM-I also outperform DBRNN. We will do our detailed analysis on these updated results. We also include the Cheng's result for DP, the result is:
> DP
> 			                  GRU
> Cheng's bi-attention:	61.29%
> To show that SLSTM retains longer memory, the closed form expression of SLSTM-I output is:
>  Y_t = \sigma (W_o I_t) \phi (\sum^t_{k=1} c_k [\prod^t_{j=k+1} \sigma(W_f I_j)] \sigma(W_i I_k) \phi(W_{in} I_k) +b) (A)
> We can pick c_k such that:
> |c_k \prod^t_{j=k+1} \sigma(W_f I_j)| > |\prod^t_{j=k+1} \sigma(W_f I_j)|	(B)
> Compare Eq. (A) to Eq. (11) (page 3), we can conclude that SLSTM-I has longer memory than LSTM. The memory capability of SLSTM-II can be proven in a similarly fashion.
> As a matter of fact, c_k plays a similarly role as the attention score in various attention models such as Vinyals et al. (2015). The impact of proceeding elements to the current output can be adjusted (either increase or decrease) by c_k.
>
> 4. Two papers for reference
> Thank you for providing these papers which we will cite as reference.
> After careful reading, we do appreciate the generalization of LSTMs, however, our proposal of SLSTM differs to those architectures by introducing a scaling factor that can enable the cell to attend to particular position of a sequence.
>
> 5. Compare Cheng's results
> We have showed Cheng's result on DP in point 3, it is outperformed by seq2seq with attention with SLSTM-I by more than 5% in accuracy.
>
> 6. Be modest in claims
> Thank you for pointing this out. We will definitely take this into consideration in our next round of revision. We do agree that there are some presentation issues that confuse readers, but we also feel that this paper introduces some noteworthy ideas that may benefit the NLP and RNN communities for their future research. But of course, we do not want to make overblown claims.

---

> ### Author Response · Authors · 2017-12-13
> **Updated Revision**
>
> Dear Reiviewer
>
> We have revised our paper according to your feedback, we have made some important revisions:
> 1. We have changed the cell name from super-long short-term memory (SLSTM) to extended-long short-term memory (ELSTM) as suggested.
> 2. The mathematical proof of the memory capability of ELSTM and the robustness of DBRNN has been added
> 3. Argument of the added number of parameters of ELSTM does not affect its practicability has been added
> 4. Experiments for sequence-to-sequence with attention and Cheng's bi-attention model have been added, the results further confirm our cell and RNN model design.
>
> best

---

### Official Review · AnonReviewer3 · 2017-11-27

**Rating:** 4
**Confidence:** 4

**Review:**

This submission first proposes a new variant of LSTM by introducing a set of independent weights for each time step, to learn longer dependency from the input. Then the submission propose a dependent bidirectional structure by using the output as input to the RNN cell to introduce the dependency of the outputs.

While LSTM do have problem to learn very long term dependency, the model proposed in this paper is very inefficient, the number of parameters are depend on the the length of sequences. Also, there is no analysis about why adding these additional weights could help the model learn better long-term dependency. In the other word, why this approach is better than attention/ self-attention? How to handle very long sequence and also, how to deal with different length? Just ignore the additional weights?

In the second part, the author argued a standard seq2seq model is vulnerable to previous erroneous predictions. But I don't understand why the DBRNN can handle it. It essentially just a multitask learning function: L = L_f + L_b + L_fb where error signal backprop to different layer directly which is not new.

The experimental results are weak. It compare with Seq2Seq model without attention. The other baseline for POS tag is from 1997.

---

> ### Author Response · Authors · 2017-12-09
> **Author Response1 (to be continued)**
>
> Thank you for your feedback, you do raise some critical points which we would like to address in the comment section, we will also reflect our answers to your concerns in the next revision of our paper.
> We do agree that there are some presentation issues that confuse readers, but we also feel that this paper introduces some noteworthy ideas that may benefit the NLP and RNN communities for their future research.
>
> 1. Efficiency of number of cell parameters
> It is true that the number of parameters of our cell design depends on the length of sequences. Before using SLSTM, one needs to decide the maximum sequence length for implementation. Such practice is also used for some popular models like sequence to sequence (Ilya Sutskever,2014) and sequence to sequence with attention (O.Vinyals, 2015). Please refer to the tensorflow implementation for bucketing here:
> https://github.com/tensorflow/nmt
> Or here from line 1145:
> https://github.com/tensorflow/tensorflow/blob/master/tensorflow/contrib/legacy_seq2seq/python/ops/seq2seq.py
> However, the memory overhead required for SLSTM is limited. In addition, SLSTM-II requires less number of parameters than LSTM for typical lengthed sequence. Please take a look at table 1 on page 5: to double the number of parameters as compare to an ordinary LSTM, the length of a sentence needs to be 4 times the size of the word embedding size and number of cells put together. That is, in the case of Ilya Sutskever,2014 with 1000 word embedding and 1000 cells, the sentence length needs to be 4x(1000+1000) = 8000! In practice, most NLP problems whose input involves sentences, the length wil be typically less than 100. As a matter of fact, the tensorflow implementation of O.Vinyals, 2015 for machine translation, the maximum sentence length is truncated up to 50. In our experiment, sequence to sequence with attention (O.Vinyals, 2015) for maximum sentence length 100 (other model settings please refer to Table 2), SLSTM-I parameters uses 75M of memory, SLSTM-II uses 69.1M, LSTM uses 71.5M, and GRU uses 65.9M. Through GPU parallization, the computational time for all four cells are almost identical with 0.4 seconds per step time on a TITAN X 1080 GPU.
> So the bottom line is: unless you are dealing with sequence length of more than hundreds of thousands, you should not worry about the memory (or the waste of it) or computational complexity issue.
>
> 2. Analysis of why SLSTM can have longer memory
> Thank you for raising this critical point! Indeed, it can be proven in a similarly fashion as we did in section 2 for LSTM. Below is the proof, and we will definitely add this part to our next revision.
> The closed form expression of SLSTM-I output is:
> Y_t = \sigma (W_o I_t) \phi (\sum^t_{k=1} c_k [\prod^t_{j=k+1} \sigma(W_f I_j)] \sigma(W_i I_k) \phi(W_{in} I_k) +b) (A)
> We can pick c_k such that:
> |c_k \prod^t_{j=k+1} \sigma(W_f I_j)| > |\prod^t_{j=k+1} \sigma(W_f I_j)|	(B)
> Compare Eq. (A) to Eq. (11) (page 3), we can conclude that SLSTM-I has longer memory than LSTM. The memory capability of SLSTM-II can be proven in a similarly fashion.
> As a matter of fact, c_k plays a similarly role as the attention score in various attention models such as Vinyals et al. (2015). The impact of proceeding elements to the current output can be adjusted (either increase or decrease) by c_k.

---

> > ### Author Response · Authors · 2017-12-09
> > **Author Response2**
> >
> > 3. Is SLSTM better than attention/self attention?
> > Thank you for reminding us to compare SLSTM with other attention models.
> > We did the experiment for sequence to sequence with attention (O.Vinyals, 2015) using different cells. The result is: seq2seq with attention with SLSTM-I outperform other cells by around 10%
> > POS, I^T_t = [X^T_t, Y^T_{t-1}]:
> > 			                              LSTM	GRU		SLSTM-I		SLSTM-II
> > seq2seq with attention	    34.10%	73.65%	        80.90%		54.53%
> > DP, I^T_t = [X^T_t, Y^T_{t-1}]:
> > 			                               LSTM	GRU		SLSTM-I		SLSTM-II
> > seq2seq with attention	    31.47%	53.70%	        66.72%		51.24%
> > It is interesting to see that seq2seq with attention with SLSTM-I also outperform DBRNN. We will do our detailed analysis on these updated results. We also include the Cheng's result for DP, the result is:
> > DP
> > 			                 GRU
> > Cheng's bi-attention:	61.29%
> > Also from the discussion in point 2, the scaling factor in SLSTMs does function like a attention score, making SLSTM inherently an attention model.
> >
> > 4. Isn't DBRNN just another so so multi-task learner?
> > The answer is yes and no.
> > DBRNN learns three different tasks: the one-directional forward, one-directional backward and bi-directional final result. In that sense, yes, DBRNN is a multi-task model.
> > However, we can show that mathematically, by doing this, DBRNN is less vulnerable to previous erroneous predictions as sequence to sequence model as follows:
> > Denote p_t as the ground truth distribution of the output at time step t, \hat{p}_t is the final prediction from the model, \hat{p}^f_t is the forward prediction, \hat{p}^b_t is the backward prediction. And we have \hat{p}_t = W^f \hat{p}^f_t + W^b \hat{p}^b_t (Eq. 36 on page 7, we just change the notation from Y to p as suggested by reviewer1). The p_t is an one-hot vector in the form of p = \mathbbm{I}(p_{tk} = k'), \forall k \in 1,...,K. Where \mathbmm{I} is indicator function, k' is the ground truth label.
> > Then the cross entropy of DBRNN is:
> > l = -\sum^K_{k=1} p_{tk} log( \hat{p}_{tk} )
> > where K is the total number of classes (size of vocabulary)
> > It can be further expressed as:
> > l = -\sum^K_{k=1} p_{tk} log( W^f_k \hat{p}^f_{tk} + W^b_k \hat{p}^b_{tk})
> >   = -log( W^f_{k'} \hat{p}^f_{tk'} + W^b_{k'} \hat{p}^b_{tk'})
> > We can pick W^f_{tk'} and W^b_{tk'} such that l < -\sum^K_{k=1} log(\hat{p}^f_{tk}) and l < -\sum^K_{k=1} log(\hat{p}^b_{tk})
> > Which means DBRNN can have less cross entropy than one-directional prediction by pooling expert opinions from that two predictions.
> >
> >  5. The experiment is weak
> > We will add the new results in point 3 to our experiment results section.

---

> ### Author Response · Authors · 2017-12-13
> **Updated Revision**
>
> Dear Reiviewer
>
> We have revised our paper according to your feedback, we have made some important revisions:
> 1. We have changed the cell name from super-long short-term memory (SLSTM) to extended-long short-term memory (ELSTM) as suggested by reviewer 2.
> 2. The mathematical proof of the memory capability of ELSTM and the robustness of DBRNN has been added
> 3. Argument of the added number of parameters of ELSTM does not affect its practicability has been added
> 4. Experiments for sequence-to-sequence with attention and Cheng's bi-attention model have been added, the results further confirm our cell and RNN model design.
>
> best

---

### Author Response · Authors · 2017-12-13
**Updated Revision**

Dear readers, please check the latest version that is revised based on the valuable feedback from the reviewers. We have change the cell name from super-long short-term memory (SLSTM) to extended-long short-term memory (ELSTM). We also include the results for sequence-to-sequence with attention model and Cheng's bi-attention model. Both results confirm our cell and RNN model design.

---

> ### Author Response · Authors · 2018-01-03
> **Other revisions**
>
> Other revisions include:
>
> 1. Clarification of SRN model in section 2
> 2. Elaboration of the GPU memory/computing speed efficiency of ELSTM at the end of section 3
> 3. Change of notations as suggested by reviewer 1
> 4. Added proof of the memory capability of ELSTM in equation 25
> 5. Added proof of  why DBRNN is robust to erroneous previous predictions in equations 32-34

---

### Decision · Program_Chairs · 2018-01-29
**ICLR 2018 Conference Acceptance Decision**

**Decision:**

Reject

**Comment:**

The reviewers of the paper are not very enthusiastic of the new model proposed, nor are they very happy with the experiments presented.   It is unclear from both the POS tagging and dependency parsing results where they stand with respect to state of the art methods that do not use RNNs.  We understand that the idea is to compare various RNN architectures, but it is surprising that the authors do not show any comparisons with other methods in the literature.  The idea of truncating sequences beyond a certain length is also a really strange choice.  Addressing the concerns of the reviewers will lead to a much stronger paper in the future.

---

> ### Author Response · Authors · 2018-01-30
> **Thank for the feedback, we appreciate the efforts ICLR committee put into the reviewing process**
>
> Dear reviewers and program chairs
>
> We appreciate your valuable feedback and will address the two concerns in future write up. It was a great experience in the process.
>
> Much thanks